# Development and Calibration of a Microfluidic, Chip-Based Sensor System for Monitoring the Physical Properties of Water Samples in Aquacultures

**DOI:** 10.3390/mi15060755

**Published:** 2024-06-04

**Authors:** Fereshteh Aliazizi, Dua Özsoylu, Soroush Bakhshi Sichani, Mehran Khorshid, Christ Glorieux, Johan Robbens, Michael J. Schöning, Patrick Wagner

**Affiliations:** 1Laboratory for Soft Matter and Biophysics ZMB, Department of Physics and Astronomy, KU Leuven, Celestijnenlaan 200 D, B-3001 Leuven, Belgium; fereshteh.aliazizi@kuleuven.be (F.A.); soroush.bakhshisichani@kuleuven.be (S.B.S.); mehran.khorshid@kuleuven.be (M.K.); christ.glorieux@kuleuven.be (C.G.); patrickhermann.wagner@kuleuven.be (P.W.); 2Institute of Nano- and Biotechnologies INB, Aachen University of Applied Sciences, Heinrich-Mussmann-Strasse 1, D-52428 Jülich, Germany; oezsoylu@fh-aachen.de; 3Cell Blue Biotech and Food Integrity, Fisheries and Food ILVO, Flanders Research Institute for Agriculture, Jacobsenstraat 1, B-8400 Oostende, Belgium; johan.robbens@ilvo-vlaanderen.be; 4Institute of Biological Information Processing (IBI-3), Research Centre Jülich, D-52425 Jülich, Germany

**Keywords:** electrical conductivity of liquids, thermometry, impedance spectroscopy, microfluidics, aquaculture, chip-based sensor setup

## Abstract

In this work, we present a compact, bifunctional chip-based sensor setup that measures the temperature and electrical conductivity of water samples, including specimens from rivers and channels, aquaculture, and the Atlantic Ocean. For conductivity measurements, we utilize the impedance amplitude recorded via interdigitated electrode structures at a single triggering frequency. The results are well in line with data obtained using a calibrated reference instrument. The new setup holds for conductivity values spanning almost two orders of magnitude (river versus ocean water) without the need for equivalent circuit modelling. Temperature measurements were performed in four-point geometry with an on-chip platinum RTD (resistance temperature detector) in the temperature range between 2 °C and 40 °C, showing no hysteresis effects between warming and cooling cycles. Although the meander was not shielded against the liquid, the temperature calibration provided equivalent results to low conductive Milli-Q and highly conductive ocean water. The sensor is therefore suitable for inline and online monitoring purposes in recirculating aquaculture systems.

## 1. Introduction

Aquaculture is a widespread, global activity aiming at providing a reliable supply of nutrient rich products of fish and other types of seafood for the world population. For Europe, the European Commission released the *Strategic guidelines for a more sustainable and competitive EU aquaculture for the period 2021 to 2030* in 2021, and its recommendations can be transferred universally to all other regions where aquaculture takes place [1]. Important points of attention are reducing the ecological footprint of aquaculture activities and, at the same time, fostering the health and welfare of the animals by gaining better control over fish diseases and infections with parasites. The spectrum of pharmaceuticals that can be applied to fish is narrow, and higher up in the food chain, residues of veterinary drugs can pose a health risk to humans. A promising route to perform aquaculture in a controlled setting are *Recirculating Aquaculture Systems* (RAS), in which the water exchange, and need for fresh water, are limited [2,3]. However, this necessitates advanced filtering techniques to remove fish excretions, and, depending on the cultivated species, the temperature, pH value, and salinity of the water need to be adjusted and stringently controlled. In recent years, there are also increasing concerns that aquaculture in general, not only in RAS, contributes to the development of antimicrobial resistance in bacterial colonies that are present in the water [4,5,6]. In this way, antimicrobial resistance can proliferate and also eventually endanger human health. For both scenarios, continuous monitoring of “environmental” aquaculture conditions by using an easy-to-handle, automatic chip-based sensor system is highly demanded.

Within the ERA-NET project ARENA (Antibiotic Resistance and Pathogenic Signature in Marine and Freshwater Aquaculture Systems), the link to the project homepage is provided in reference [7], we are developing a modular sensor system that is adapted for RAS facilities. The two central modules will measure (i) the temperature and electrical conductivity of RAS water, and (ii) the concentration of potentially pathogenic bacteria in a species-selective way. In the present work, we will address the chip-based temperature and conductivity sensor setup; the bacterial sensor is still under optimization, and the details will be published elsewhere. All sensors should be suited for continuous monitoring, even under the harsh conditions of exposure to seawater, and the output data should be in an electronic format for data logging and remote control.

While capacitively coupled contactless conductivity detection (C4D) devices are the state of the art for measuring the electrical conductivity of liquids [8,9], we will employ impedimetric sensing with interdigitated electrodes in the present work for several reasons: First, impedance spectroscopy is a mature technique, and the fundamentals are documented, e.g., in references [10,11]. A tutorial on the application of impedance analysis in various scientific fields was published recently by Lazanas and Prodromidis [12]; an up-to-date overview on capacitive field-effect devices for the detection of charged molecules can be found in reference [13]. Furthermore, multiplexed impedance analyzers currently come in a very compact size and at a reasonably low price. Second, once impedance analysis, combined with interdigitated electrodes, is established for temperature sensing, the basic concept is transferable to bio-detection. Using appropriate bioreceptors, impedance spectroscopy has been used for bacterial detection in complex samples down to low concentrations, shown in reference [14], and the synthesis protocol for whole-cell bacterial receptors has been documented in [15]. Impedance analysis, when combined with molecularly imprinted polymers as receptors, has also been successfully employed to detect trace-level concentrations of antibiotics, e.g., Chloramphenicol and Norfloxacin, in water and aquaculture samples [16,17]. Hence, opting for conductivity measurements based on impedance spectroscopy, the sensor will be ready for subsequent upgrades to detect targets such as bacteria and antimicrobials, which both have a high relevance in the context of aquaculture and antimicrobial resistance [4,5,6].

Regarding temperature sensing, we developed planar meander structures, deposited together with the interdigitated electrode structure on the same sensor chip. The material of choice is platinum, which is not only corrosion-resistant, but also has a high temperature coefficient of resistance (TCR), with a literature value of 3.93 × 10^−3^ K^−1^ [18]. The TCR, describing the resistance change in a conductor with increasing temperature, plays a central role in thermal conductivity measurements based on the so-called 3-omega technique [19]. The 3ω principle was, e.g., used to measure the impact of organic layers on the thermal interface resistance between metal microwires and phosphate buffer [20], as well as to study the formation of bacterial biofilms on on-chip platinum meanders [21]. Such biofilms play an important, typically unwanted role in food industry [22], and hence, the temperature-sensing meander can, in principle, be used for biofilm monitoring in aquaculture as an add-on application.

## 2. Materials and Methods

### 2.1. Sensing Elements and Microfluidic Channel Device

The sensor for measuring the electrical conductivity via the impedance signals consists of five sets of interdigitated electrodes (IDES) fabricated by photolithographic patterning on a glass chip (width 25.5 mm, length 67.5 mm, and ≈0.5 mm thickness), shown in Figure 1 (the glass chip holds five IDES structures together with one temperature sensor as the meander on the left). Before sensor chip fabrication, firstly, CAD software (KLayout, version number 0.27.4, Munich, Germany) was used to design the geometry of the sensing elements (electrodes). For the conductivity (impedance) sensor, the geometry of the interdigitated electrodes was adjusted as follows: the finger width as 5 µm, the spacing between the fingers as 5 µm, the length of the fingers as 6.76 mm, and a total number of 500 of the fingers (each IDES structure contains 2 × 250 individual strips, and the size of each IDES structure is 5.0 mm by 8.0 mm). For the temperature sensor, the total length and width of the meander-type structure were designed as 30.5 mm and 300 µm, respectively, with a resistance of *R* ≈ 140 Ω, measured in four-point geometry at 25 °C.

For the fabrication of the sensor chip, the electrode patterns on a glass substrate were obtained using direct laser writing lithography. An image reversal photoresist (AZ 5214E [JP], diluted 1:0.476, Microchemicals GmbH, Ulm, Germany) was spin-coated (4000 RPM, 30 s) on the glass substrate (glass wafer, 0.5 mm thick, Borofloat 33, SIEGERT WAFER GmbH, Aachen, Germany). After a soft bake at 105 °C for 1.5 min, laser writing was performed using a direct laser writer device (PicoMaster 150, Raith GmbH, Dortmund, Germany) by applying an exposure energy of 35.96 mJ/cm^2^ (405 nm laser source, spot size: 550 nm, step resolution: 275 nm). Afterwards, a flood exposure of 900 mJ/cm^2^ was applied after a reversal bake at 120 °C. Then, the electrode patterns were obtained after developing with tetramethyl-ammonium hydroxide (TMAH, 2.38% in H_2_O, AZ 726MIF, Microchemicals GmbH, Ulm, Germany) for 1 min. For the metal deposition on the achieved patterns, the physical vapor deposition technique (electron beam evaporation) was used. For this, the wafer with patterns was deposited with a platinum layer (100 nm) after coating with titanium as an adhesion layer (10 nm). A lift-off process was utilized using dimethyl sulfoxide (Micro D350, Microchemicals GmbH, Ulm, Germany) for 2 h at 60 °C. After the lift-off process, the wafer was diced to obtain the sensor chips with a size of 25.5 mm (width) and 67.5 mm (length).

Figure 1b shows a zoom-in photograph on the left part of Figure 1a of the temperature sensor (top), and the first IDES structure (bottom) for conductivity studies. For the actual measurements, we constructed a microfluidic channel (5.0 mm wide, 50 mm long) by bonding a top lid onto the glass chip using a Ibidi sticky-Slide I^0.8^ Luer adhesive tape (ibidi GmbH, Gräfeling, Germany), resulting in an inner channel height of 0.8 μm, and a channel volume of (200 ± 12.5) μL, according to the specifications by the manufacturer. The top lid has an inlet and outlet at opposite ends of the channel, where the sample can be administered via Teflon tubing connected to a syringe (see Figure 1c). Figure 1d depicts the channel with the electrical connectors for temperature and impedance measurements.

### 2.2. Impedance Measurements and Reference Electrolytes

The general idea of the proposed approach is to determine the conductivity of water samples from their impedance data, obtained with the IDES structures. For measuring the impedance spectra, we utilized an Autolab PGSTAT 302N impedance analyzer (Metrohm, Herisau, Switzerland). The AC (alternating current) voltage amplitude was 65 mV under open circuit conditions, and the measured frequencies ranged from 10 Hz up to 500 kHz, with five frequencies per decade, spaced equidistantly at a logarithmic scale. Measuring an entire impedance spectrum took 60 s. To establish the relationship between the electrical conductivity and the impedance signals, we studied a series of reference electrolytes, shown in Table 1, with a known concentration and composition of ions.

The conductivity and pH values of these electrolytes were measured with a calibrated pH and conductivity meter SevenCompact Duo (Mettler Toledo N.V., Zaventem, Belgium), at room temperature. This instrument, SevenCompact Duo, is considered as an independent reference instrument throughout this study. Regarding the reference electrolytes, we prepared a dilution series of phosphate-buffered saline (PBS) solution, starting from 10 × PBS and resulting in the concentrations given in Table 1. The 10 × PBS was obtained by dissolving three PBS tablets (Sigma-Aldrich, Saint Louis, MO, USA) in 60 mL of Milli-Q water. The dilutions were then prepared from the original stock solution (10 × PBS) using Milli-Q water. The ion concentrations in 1 × PBS are 137 mM NaCl, 10 mM Na_2_HPO_4_, and 2.7 mM KCl.

### 2.3. Temperature Calibration Measurements

For the temperature measurements, the microfluidic channel with the platinum RTD meander was placed into a temperature-controlled incubator (model R-TH-50, Labtech Instrument Co., Ltd., Dongguan, China). The temperature setpoints were 2, 5, 10, 15, 20, 25, 30, 35, and 40 °C, and each temperature was kept for 30 min to establish a stable thermal equilibrium between the channel and the surrounding incubator. The meander resistance was measured at each temperature in four-wire geometry using a Hewlett Packard 34401 multimeter (Palo Alto, CA, USA) kept outside the incubator at room temperature. The absolute temperatures inside the incubator were also verified using a calibrated Pt100 resistor (Labfacility Ltd., Bognor Regis, UK), positioned close to the microfluidic device, which was connected to a second, technically identical, multimeter. To check for potential hysteresis effects, measurements were also performed in the order from high to low temperature, i.e., from 40 °C down to 2 °C. All measurements were conducted with different fillings of the microfluidic channel, as follows: empty (air filled), filled with Milli-Q water, with 1 × PBS, and with sea water; see Table 2, below, for more information.

For each new medium, the channel was flushed with 20 mL of the new medium, corresponding to 100 times the channel volume. While the sensor will eventually be used with sea water, and we studied the different channel fillings in order to assess whether the salinity of the medium creates a parallel current path. If this is the case, it will affect the measured resistance of the meander or its temperature coefficient of resistance (TCR). In the case that there is no parasitic current, the meander can be used as it is shown in Figure 1a,b, above.

### 2.4. Water Samples under Study

The water samples were chosen along the intended line of applications in aquaculture, and can be divided into the following three major groups, shown in Table 2: The samples # 1–3 stem from a river and two channels in Belgium; here, we do not expect major differences in their pH value and comparatively low conductivity. These specimens were taken at moments without rainfall during the preceding days to avoid dilution, and, between sampling and the actual measurements, the samples were stored refrigerated at 4 °C (which is also the case for the other specimens). By measuring the pH and conductivity of several samples before and after storage, we found consistent values, meaning that storage did not affect the physical properties of the specimens. Prior to storage, the samples were filtered with syringe filters, featuring a nylon membrane with 0.2 μm mesh size (VWR, Haasrode, Belgium), to remove potential microorganisms. The samples # 4–8 were collected directly from the recirculating aquaculture basins of the ILVO institute in Ostend (Belgium). Essentially, these specimens consist of sea water with the following two exceptions: the samples # 4 and # 7, in which mullet and sole are, respectively, cultivated, are brackish water, i.e., a mixture of sea and fresh water with medium-to-high conductivity. The third group, samples # 9 and 10, consists of sea water from two different locations, and we mention that the Eastern Scheldt is also connected to the North Atlantic. Still, the salinity and resulting (high) electrical conductivity might display minor differences. The pH data, obtained using the SevenCompact Duo instrument, are all at the slightly alkaline side, with the exception of the samples from the sea bream basin (samples # 5 and 6). Their slightly acid pH seems related to microbial activity, and was confirmed by independent reference measurements at the ILVO institute. The pH of the ocean waters (samples # 9 and 10), which were taken at coastal regions, are slightly below the pH found in the open sea, which is between pH 8.04 and 8.13, depending on the season; see reference [23] for more information.

## 3. Results and Discussion

### 3.1. Electrical Conductivity of the Reference Electrolytes

To study first the relationship between the salt concentration of an electrolyte and the resulting electrical conductivity, we measured the conductivity *σ* of the reference electrolytes, provided in Table 1. For each of these samples, the salt concentration is exactly known, which is not the case with the river, aquaculture, and ocean samples introduced in Table 2. Figure 2 shows the conductivity data of the electrolytes, measured using the SevenCompact Duo conductivity meter at room temperature, as a function of the PBS concentration ranging from pure Milli-Q water to 10 × PBS. To include all data points, we use a double logarithmic scale that has a good agreement (*R*^2^ = 0.999) with the empirical Kohlrausch formula given in Equation (1) [24,25], as follows:(1)$$\sigma \;\left(c\right)=c\;\cdot \;\left(A_{1}-A_{2}\;\sqrt{c}\right)$$

Here, *c* is the salt concentration with a nominal value of *c* = 1 for 1 × PBS, corresponding to 150 mM based on the composition of 1 × PBS. The fit parameters *A*_1_ and *A*_2_ were determined using OriginLab^®^ software (OriginPro 2023b), and their numerical values are *A*_1_ = (1.933 ± 0.014) × 10^4^ and *A*_2_ = (2.213 ± 0.047) × 10^3^, yielding the conductivity in units of μS cm^−1^. Hence, when measuring the conductivity of an unknown sample, it is possible to estimate its salt concentration *c*, relative to 1 × PBS, either by solving Equation (1) for *c*, or by using Figure 2 as a look-up table. Furthermore, it is also possible to derive from *A*_1_ the averaged molar conductivity of the ions present in PBS. Deviations from the Kohlrausch fit occur for salt concentrations below 10^−3^ × PBS and *σ* < 10 μS cm^−1^, respectively, because even distilled water has a non-zero conductivity due to the presence of H^+^ and OH^−^ ions, as well as the uptake of CO_2_ from the environment [26,27].

Ignoring these effects, the Kohlrausch formula (Equation (1)) indicates that the conductivity is a linear function of the salt concentration in the regime of low concentrations (*c* → 0). However, for high concentrations, Equation (1) predicts a sub-linear increase in σ with increasing electrolyte concentrations; this is due to the formation of solvated, tightly bound anion-cation couples that are electrically neutral and can, therefore, not contribute to the electric transport. In conclusion, Equation (1) is in good agreement with the data over almost four orders of magnitude in absolute conductivity values.

### 3.2. Impedance Spectra of Water Samples and Reference Electrolytes with Chip-Based Sensor Setup

Next, we studied the impedance spectra with IDES electrodes in the microfluidic channel at 19 °C in the frequency regime from 100 Hz to 500 kHz, shown in Figure 3, for the reference electrolytes (Table 1) and water samples (Table 2). For each sample, the impedance spectrum was measured once, and the width of the noise band around each data point of Figure 3 is below the size of the symbols. Hereby, we used the same chip (see Figure 1) for all samples to ensure consistency of the data and, after measuring one sample, we injected at least 5 mL (25 times the channel volume) of the following type of sample to guarantee a complete replacement of the previous fluid.

The data are presented as Bode plots in Figure 3a for the reference electrolytes (which includes selected specimens of low, intermediate, and high conductivity to avoid overlapping of the data), and all water samples in Figure 3b. For both diagrams, the impedance amplitude can be mainly described by a capacitively dominated behavior in the frequency range between 100 Hz and around 1 kHz (with the exception for the low conductivity solutions in Figure 3a). The capacitive behavior is also supported by the phase angles between −78.3° and −81.8° at 100 Hz; the exact value is sample dependent. While at higher frequencies (≥10^4^ Hz), the chip-based conductivity sensor shows resistive behavior, whereas the impedance amplitude decreases with higher analyte conductivity values. Based on these results, we defined a selected frequency, *f**, to establish a one-to-one correlation with the reference lab instrument, and to guarantee a quantitative calculation by the chip-based IDES setup (see also Section 3.3). For the given frequency of *f**, the phases’ angles range from −7.0° to −12°, indicating a mainly resistive behavior.

### 3.3. Single-Frequency Algorithm for Conductivity Measurements with the IDES Sensor

The impedance spectrum, obtained for a given sample, allows us to, in principle, to derive the electrical resistance of the fluid and its conductivity, respectively. This procedure involves the modelling of the data to an electrically equivalent circuit, and the outcome depends on certain assumptions regarding the number and type of the elements included in the circuit. To work in a way that is assumption free and avoids modelling, we have opted for a different strategy, as follows: We search for a frequency *f** (or a set of frequencies), at which there is a one-to-one correspondence between the measured impedance amplitude |*Z*(*f**)| and the actual conductivity *σ* of the sample. This way, it will be sufficient to limit the measurements to this single frequency, which speeds up the measurements and can be implemented with a simplified hardware that possibly does not even require an impedance analyzer.

We screened multiple frequencies for this purpose, focusing especially on frequencies above 10 kHz, where the data will not be dominated by the capacitive effects upon the interface between the liquid and the IDES electrodes. Amongst all probed frequencies, *f** = 46.933 kHz turned out to provide the best results. Figure 4 presents the resulting calibration curve, based on the fit function of Equation (2), which shows the impedance amplitude of the calibration electrolytes as a function of their conductivity, measured using the reference lab instrument. Figure 4 covers the conductivity range between 100 and 10^5^ μS cm^−1^, i.e., the relevant range for the freshwater, seawater, and aquaculture samples.

**Figure 4 micromachines-15-00755-f004:**
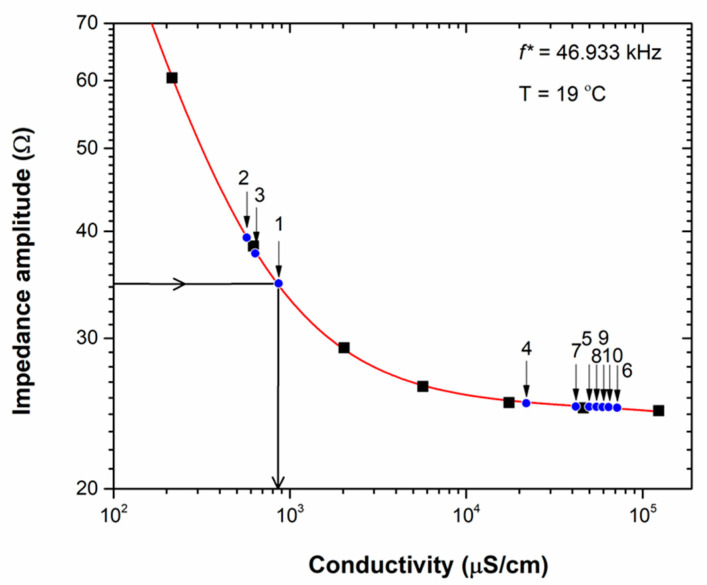
The calibration curve (red solid line), based on the fit function of Equation (2), renders the relationship between the impedance amplitude of the chip-based IDES structures and the measured conductivity of the reference electrolytes (the data points are given as black-box symbols). The numbers indicate the position of all water samples (blue dot symbols) in terms of their impedance amplitude, and each impedance value refers to a unique conductivity value. This is shown exemplarily for sample # 1: the |*Z*| amplitude defines the position of sample # 1 on the calibration curve, and this position relates to the conductivity of the samples as indicated by arrows. The samples fall roughly into two groups: # 1–3 are freshwater specimens, and samples # 4–10 are seawater and aquaculture specimens. The conductivity values of the water samples, derived from the calibration plot, agree closely with the conductivities determined using the reference lab instrument (Table 3).

The calibration curve in Figure 4 takes all reference electrolytes that have an ionic strength in the range from 10^−3^ × PBS up to 10 × PBS into account. Electrolytes with a lower salinity were ignored because their conductivity is far below the conductivity of the water samples of interest. The fit function is empirical, and given by the following equation:(2)$$\left|Z\right|\left(f^{*}\right)=\frac{A+B\cdot \sigma }{1+C\cdot \sigma +D\cdot \sigma^{2}}$$

Here, |*Z*| is the impedance amplitude at the selected frequency, *f** = 46.933 kHz in Ohm (Ω) units, and *σ* is the electrolyte conductivity in units of μS cm^−1^. The fit parameters were obtained using OriginLab^®^ software, and have the numerical values of *A* = 350.30, *B* = 0.94, *C* = 0.037, and *D* = 4.75 × 10^−9^; the goodness factor is *R*^2^ = 0.99999. For all water samples, i.e., # 1–10, we took the impedance amplitude of *f** = 46.933 kHz from Figure 3b, and derived the related conductivity *σ* graphically from the calibration plot of Figure 4. Alternatively, one can solve Equation (2) numerically for *σ* when |*Z*| is given. The results, as follows, are summarized in Table 3: there is an excellent agreement between the conductivities derived from the impedance amplitude at the frequency *f** with the chip-based IDES structures and measurements on the ten different water samples using the reference lab instrument. In conclusion, the algorithm, based on the impedance at a single frequency, provides accurate conductivity data in the broad range from river to ocean water.

### 3.4. Calibration of the Chip-Based Platinum RTD

Figure 5 shows the resistance data of the platinum RTD for temperatures varying from 2 °C to 40 °C. The resistance values represent the average obtained from monitoring the resistance signal for a period of 10 min, i.e., between 20 and 30 min after setting the nominal temperature. The considered temperature range is broader than the typical water temperatures of the Atlantic Ocean near the Belgian coast, which neither drop below 4 °C nor exceed 21 °C [28]. With indoor RAS systems, the water temperature is typically around room temperature, with seasonal fluctuations. The resistance values change on a strictly linear basis with temperature, and for each temperature, the resistance was measured twice, at the following times: during the heating steps from 2 °C to 40 °C, and during the cooling steps from 40 °C to 2 °C. Within the resolution of the Hewlett Packard 34401 multimeter, there was no measurable difference between the two resistance values of the sensor chip at a given temperature, and this holds for the four different media inside the microfluidic channel (air, Milli-Q water: MQ, 1 × PBS, and sea water from Ostend harbor).

All resistance values at 20 °C, shown in Table 4, fall within a narrow range of 139.5 ± 0.3 Ω, meaning that even the highly conductive sea water sample (σ ≈ 50 mS cm^−1^) does not act as a parallel conductance channel when measuring the meander-type platinum RTD. Therefore, the chip-based temperature sensor can be used as is, without the need for additional passivation layers. For comparison, McAdams et al. studied the charge-transfer resistance *R_CT_* between gold electrodes and phosphate buffer, resulting in values in the MΩ range, which, similarly, seems to be the case with the platinum sea water interfaces in our present study [29]. Here, we mention that the conductivity of salt solutions increases substantially with increasing temperature [30], but this effect is not seen in Figure 5, supporting the argument of a high electrical interface resistance between the meander and the fluids.

The sensitivity of the RTD meander (with respect to temperature changes) is determined by its resistive temperature coefficient, for which the value is 2.30 × 10^−3^ K^−1^. This is lower than the literature value of 3.93 × 10^−3^ K^−1^ for platinum at 20 °C [18], which we attribute to impurities, stress, and grain boundaries in the meander, deposited by physical vapor deposition as a thin-film sensor. Still, the chip-based RTD is sufficiently sensitive to temperature changes. Any multimeter, which does not need to be a high-end instrument, can detect resistance changes by 10 mΩ, which translates in our case to a temperature change of ±0.03 °C. For aquaculture purposes, it is generally enough to know water temperatures with a precision of 0.5 °C.

## 4. Conclusions

In the present work, we have provided a progress update on the development of a sensor system that will eventually serve in aquaculture applications to continuously measure important water parameters *inline*, directly within fish basins. Specifically, for recirculating aquaculture systems (RAS), measurements can also be performed *online* by installing the sensor in a bypass to the pumping system. Currently, the sensor setup is able to measure the temperature of water samples and their electrical conductivity, which is related to the salt concentration. The sensor system is open to upgrades for measuring the pH, and possibly the oxygen concentration p(O_2_), as well as bacterial contaminations. For bacterial detection, the interdigitated electrode structures that are not yet in use will be functionalized with species-selective receptors, such as surface-imprinted polymers.

At the methodological side, we have developed an algorithm that allows us to determine the electrical conductivity of a water sample directly from the impedance amplitude at a single trigger frequency, without the need for equivalent circuit modelling. The conductivity values determined this way show a very good agreement with the absolute conductivities measured using a calibrated, high-end reference lab instrument. This holds for a broad spectrum of water specimens that includes surface waters from rivers and channels, sea water from the Atlantic Ocean and the Eastern Scheldt, as well as samples from a RAS-type aquaculture facility. Amongst the RAS samples, the sensor can distinguish clearly between sea water and brackish water. The temperature sensor, designed as a platinum RTD meander that is on the same chip as the interdigitated electrode structures, shows no hysteresis effects between warming and cooling cycles; its resolution, with respect to temperature changes, is better than 0.03 °C. Importantly, the chip-based temperature sensor does not show corrosion in contact with sea water from the Atlantic Ocean, and the high electrical conductivity of sea water does not affect the temperature calibration in comparison to measurements under dry conditions.

## Figures and Tables

**Figure 1 micromachines-15-00755-f001:**
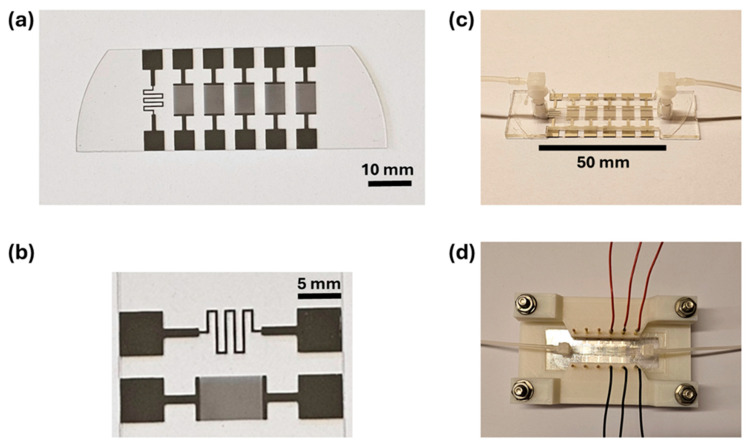
(**a**) Glass chip with a meander-type temperature sensor made of platinum, and five interdigital electrode structures’ IDES; panel (**b**) is a zoom-in. One IDES structure is sufficient to determine the electrical conductivity of the medium, while the four others can be functionalized, e.g., with bioreceptors for bacterial detection. (**c**) Mounted microfluidic channel with an inlet and outlet for fluids. (**d**) The channel with electrical connections for temperature and impedance measurements.

**Figure 2 micromachines-15-00755-f002:**
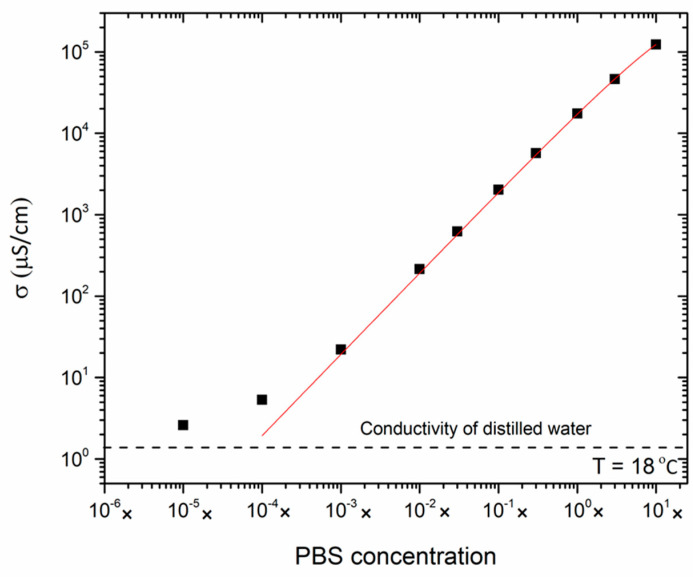
Conductivity of the electrolyte solution as a function of the PBS concentration at room temperature. The fit function (red line) is based on the Kohlrausch formula in Equation (1), with *R*^2^ = 0.999. Deviations from the fit occur at the lowest salt concentrations, where the conductivities are close to the limit of distilled water (horizontal dashed line); see references [26,27] for reference values. Each data point is the average of three independent measurements, and the error bars are smaller than the symbol size.

**Figure 3 micromachines-15-00755-f003:**
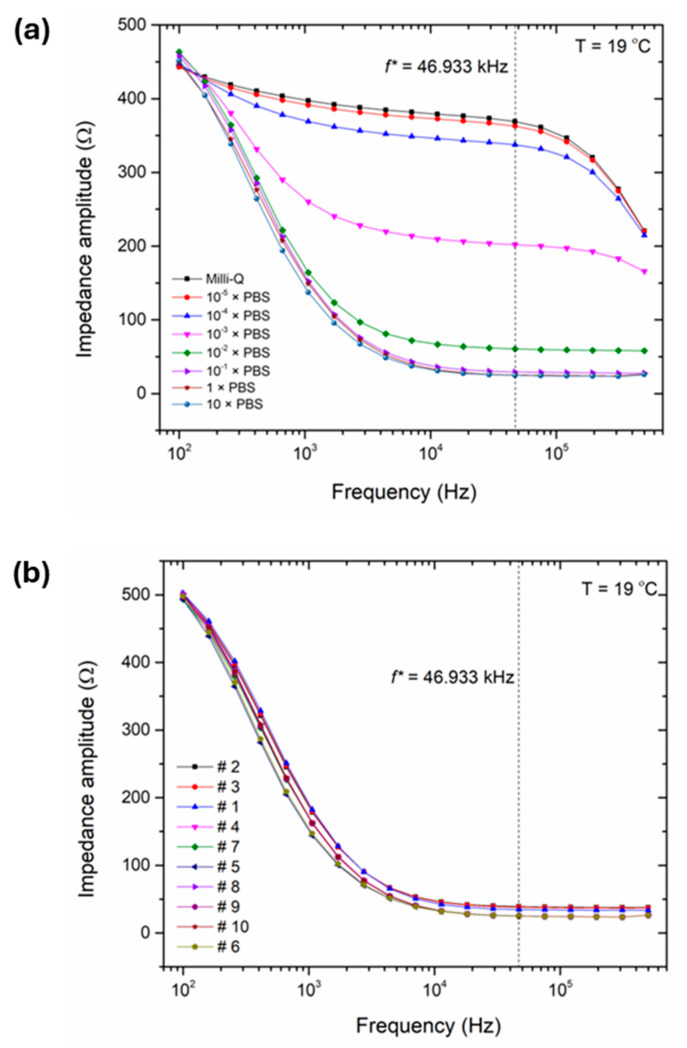
(**a**) Impedance spectra at 19 °C (Bode plots: 100 Hz–500 kHz) of selected electrolyte samples of Table 1 used for calibrating the IDES-based conductivity sensor. Panel (**b**) shows the corresponding spectra of the water samples of Table 2 within the same frequency range. The order from samples # 2 to # 6 corresponds with the decrease of the impedance amplitude and an increase in the conductivity (*σ* is lowest for sample # 2 and highest for sample # 6), respectively. For comparison, we refer also to Figure 4. The dashed, vertical line indicates the selected frequency, *f** = 46.933 kHz, used to establish a one-to-one correlation between the impedance amplitude and the conductivity as determined using the reference lab instrument (see also Section 3.3).

**Figure 5 micromachines-15-00755-f005:**
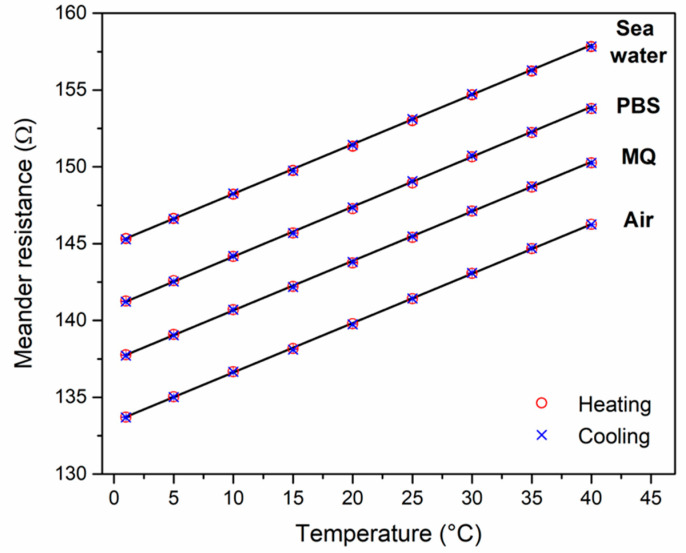
Resistance of the temperature-sensitive platinum RTD, measured in four-point geometry in an incubator at temperatures from 2 °C up to 40 °C (heating, red symbols), and back to 2 °C (cooling, blue symbols). All data points have an uncertainty that is much smaller than the actual symbol size. The data for the empty microfluidic channel (air filled) are the measured values. The resistance data for Milli-Q water (MQ) have been shifted vertically by +4.0 Ω for clarity; the data for 1 × PBS (PBS) are shifted by +8 Ω; and the data for sea water (sample # 9) are shifted by +12 Ω. For each incubator temperature, the resistance values obtained during heating and during cooling cycles coincide. The solid, parallel lines are linear fits to the data obtained with increasing temperatures, indicating that the TCR is independent of the medium inside the microfluidic channel.

**Table 1 micromachines-15-00755-t001:** Reference electrolytes used for calibrating the chip-based impedimetric conductivity sensor. All data were obtained using the reference instrument Mettler-Toledo SevenCompact Duo. The temperature of all fluids was between 18.6 and 19.8 °C, and we utilized the temperature-compensation mode of the instrument. All data are averages of three independent measurements, and the uncertainties are the standard deviations.

PBS Concentration	pH Value	Conductivity σ (μS cm^−1^)
10.0×	6.78 ± 0.01	(1.233 ± 0.002) × 10^5^
3.0×	7.19 ± 0.01	(4.624 ± 0.009) × 10^4^
1.0×	7.43 ± 0.01	(1.749 ± 0.004) × 10^4^
0.3×	7.56 ± 0.01	(5.679 ± 0.009) × 10^3^
0.1×	7.55 ± 0.01	(2.032 ± 0.008) × 10^3^
3.0 × 10^−2^×	7.42 ± 0.01	621.5 ± 1.1
1.0 × 10^−2^×	6.93 ± 0.01	215.1 ± 0.2
1.0 × 10^−3^×	6.01 ± 0.00	22.17 ± 0.12
1.0 × 10^−4^×	5.84 ± 0.02	5.340 ± 0.013
1.0 × 10^−5^×	5.73 ± 0.00	2.614 ± 0.009
Milli-Q water	6.08 ± 0.07	2.211 ± 0.034

**Table 2 micromachines-15-00755-t002:** Overview of the water samples analyzed with respect to their pH value and their electrical conductivity *σ*. For the conductivity data, see Section 3.3. Except for sample # 9, all samples were sourced in Belgium; the RAS samples were collected at the recirculating aquaculture facility of the ILVO institute in Ostend. The pH values are averages of three independent measurements obtained using the reference instrument (SevenCompact Duo), and the uncertainty is the standard deviation.

Sample Type	Sample Number	Place, Description	pH Value
Surface waters	# 1	Dyle river (Leuven, Arenberg castle)	8.08 ± 0.01
	# 2	Leuven-Dyle channel (Mechelen, Stuyvenbergvaart)	7.89 ± 0.01
	# 3	Brussels-Scheldt sea channel (Grimbergen, Verbrande Brug)	7.84 ± 0.01
Recirculating aquaculture systems RAS	# 4	Mullet (*Mugil cephalus*), cultivated in brackish water	7.35 ± 0.01
	# 5	Sea bream (*Abramis brama*)	5.76 ± 0.01
	# 6	Sea bream (*Abramis brama*), newly placed filters	5.82 ± 0.01
	# 7	Sole (*Solea solea*), lightly brackish water	8.11 ± 0.01
	# 8	Gray shrimp (*Crangon crangon*)	7.54 ± 0.01
Ocean waters	# 9	Sea water from Eastern Scheldt (The Netherlands)	7.97 ± 0.01
	# 10	Atlantic ocean water from Ostend Harbor	7.86 ± 0.01

**Table 3 micromachines-15-00755-t003:** Conductivity values of all water samples, comparing the conductivities measured using the reference lab instrument to those obtained with the IDES-based impedance value at the frequency *f**. The relative deviation between the conductivities obtained with the impedance algorithm and those measured using the reference lab instrument is always below 4%; in seven cases it is even below 0.5%. The conductivity values determined using the reference instrument are averages of three independent measurements, and the uncertainties are the standard deviations. The conductivity data derived from the impedimetric sensor are based on the fit function Equation (2), plotted in Figure 4. The uncertainties here emerge from the noise band around the impedance amplitudes at *f**.

Sample Number	Conductivity σ (μS cm^−1^) Reference Instrument	Conductivity σ (μS cm^−1^) Impedimetric Sensor	Relative Deviation
# 1	855.8 ± 0.5	854 ± 77	−0.2%
# 2	574.6 ± 0.8	589 ± 48	+2.5%
# 3	616.2 ± 0.9	626 ± 44	+1.6%
# 4	(21.40 ± 0.03) × 10^3^	(21.38 ± 0.64) × 10^3^	−0.1%
# 5	(48.80 ± 0.05) × 10^3^	(46.89 ± 1.39) × 10^3^	−3.9%
# 6	(50.86 ± 0.08) × 10^3^	(50.68 ± 2.70) × 10^3^	−0.4%
# 7	(41.18 ± 0.07) × 10^3^	(41.20 ± 1.08) × 10^3^	<+0.1%
# 8	(49.21 ± 0.06) × 10^3^	(49.38 ± 1.23) × 10^3^	+0.3%
# 9	(49.62 ± 0.07) × 10^3^	(49.81 ± 1.56) × 10^3^	+0.4%
# 10	(50.20 ± 0.03) × 10^3^	(50.24 ± 2.86) × 10^3^	<+0.1%

**Table 4 micromachines-15-00755-t004:** Resistance and TCR values of the on-chip platinum RTD meander for different media inside the microfluidic channel. With the conductive liquids, 1 × PBS and sea water, the resistance is only marginally lower than for air and Milli-Q, and the relative difference is less than 0.4%. The goodness factor *R*^2^ indicates that the TCR can be considered constant within the temperature range from 2 to 40 °C. The uncertainty of the resistance data represent the width of the noise band over an observation period of 10 min.

Medium	Resistance at 20 °C (Ω)	TCR (10^−3^ K^−1^)	*R*^2^ Linear Fit
Air	139.779 ± 0.002	2.303	0.9999
Milli-Q	139.741 ± 0.001	2.273	0.9996
1 × PBS	139.274 ± 0.004	2.301	0.9998
Sea water (# 9)	139.329 ± 0.005	2.303	0.9999

## Data Availability

Data are available upon on a written request to co-author Patrick Wagner, email: patrickhermann.wagner@kuleuven.be.

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
