# Peer review of "Development and Calibration of a Microfluidic, Chip-Based Sensor System for Monitoring the Physical Properties of Water Samples in Aquacultures"

_micromachines, 2024, doi:10.3390/mi15060755_

Round 1
Reviewer 1 Report
Comments and Suggestions for Authors
The paper present a compact, bifunctional chip-based sensor setup that measures the temperature and electrical conductivity of water samples, including specimens from rivers and channels, aquaculture, and the Atlantic Ocean. Articles can be relatively innovative and need to address the following issues before they can be published.
1. The paper has too little data for the characterization of the chip and lacks specific chip structure parameters etc. The chip fabrication process needs a detailed process.
2. The working principle of the chip needs to be specified.
Comments on the Quality of English LanguageNo.
Author Response
Modifications and amendments to the text, based on the remarks by both Reviewers, are highlighted with a yellow background colour.
Minor textual changes, unrelated to the Reviewers’ comments, are indicated in light-blue colour. This holds also for information that the Reviewers did possibly not notice.
Reviewer 1
General comment: The paper present a compact, bifunctional chip-based sensor setup that measures the temperature and electrical conductivity of water samples, including specimens from rivers and channels, aquaculture, and the Atlantic Ocean. Articles can be relatively innovative and need to address the following issues before they can be published.
Comment 1: The paper has too little data for the characterization of the chip and lacks specific chip structure parameters etc. The chip fabrication process needs a detailed process.
Reply: Thank you for your hint. Correspondingly, more data have been added to the manuscript to describe the fabrication process of the sensor chip in more detail (see pages 3,4 in Section 2.1).
Comment 2: The working principle of the chip needs to be specified.
Reply: For the temperature measurement, the meander-type temperature sensor is working as RTD. For the conductivity (impedance) studies, the IDES structure enables the measurement of electrical conductivity, as described e.g., in [12].
The authors would like to thank the reviewer for the valuable comments and suggestions to improve the quality of the manuscript.
Reviewer 2 Report
Comments and Suggestions for Authors
Well thought out article. Arguments are easy to follow and understand. I have a few comments about the article.
1. Materials and methods - a key tenet of journals is that the work should be repeatable. There is insufficient information in this section to determine the validity of the processing procedures. There is a lack of information on the fabrication of the sensors and/or the lack of a reference for it in 2.1.
2. In 2.3, it is mentioned that temperatures were kept at 30mins to establish equilibrum. There is not discussion of how this 30mins is establish. Is it experimentally or theoretically determined?
3. In 2.3, there is no discussion on how cross contamination between samples, if applicable, is eliminated when filling the channels with water/PBS/sea water assuming they are all measured using the same chip. If different chips were used, how did you ensure calibration between the various devices?
4. In 2.4, it is mentioned that the samples were taken from areas of intended applications in aquaculture however without a further explaination on what they are, the sample looks seemingly random as its presented. It will be beneficial to the overall understanding for the reader to understand why these samples were picked. For example why was surface water included in the sample mix and how it links to aquaculture as indicated in the title.
5. In 2.4, samples were stored at 4C. Does this apply to all samples and how long are they stored on average before testing? Is the sample tested at room temp so the samples will need to be brought up to room temp prior? Does this degrades the samples? Have test been done to check the difference between the results from a new sample and a stored sample? With stored samples, with the cooling and bringing it back to temp, evapouration and condensation occurs and this affects certain parameters especially in sea water which will affect conductivity. This needs to be discussed further.
6. In 2.3, we know that conductivity measurements are strongly affected by the temperature of the sample. How is this controlled while taking measurements? What is the sample size and quantity of each sample and how many test per sample that informs your graphs.
7. Statically information should contain error bars with standard deviations to representations of the variability of data and used on graphs to indicate the error or uncertainty in a reported measurement. This should extend to all plots presented in the article.
8. 3.3 should really be pulled out and discussed in a separate chapter as it discusses your method. It is kind of lost as a sub-section under results.
9. Lines 422-427 and 438-442 reads identical. Only difference is the mentioned of RTD mender in 422 and just mender in 438. All other details are identical. Could the authors please clarify or make more of a distinction between the 2.
10. Figure 4b way of presenting the sample impedence is somewhat confusing. Numbers not in order just looks weird unless there is a specific rationale for this way of presentation. If so this needs to be discussed.
Author Response
Modifications and amendments to the text, based on the remarks by both Reviewers, are highlighted with a yellow background colour.
Minor textual changes, unrelated to the Reviewers’ comments, are indicated in light-blue colour. This holds also for information that the Reviewers did possibly not notice.
Reviewer 2
General comment: Well thought out article. Arguments are easy to follow and undershitand. I have a few comments about the article.
Comment 1: Materials and methods - a key tenet of journals is that the work should be repeatable. There is insufficient information in this section to determine the validity of the processing procedures. There is a lack of information on the fabrication of the sensors and/or the lack of a reference for it in 2.1.
Reply: Thank you for your hint. Correspondingly, more data have been added to the manuscript to describe the fabrication process of the sensor chip in more detail (see pages 3,4 in Section 2.1).
Comment 2: In 2.3, it is mentioned that temperatures were kept at 30mins to establish equilibrum. There is not discussion of how this 30mins is establish. Is it experimentally or theoretically determined?
Reply: The incubator (utilized in this experiment) was new and had not been used before, therefore we could not anticipate the duration to reach thermal equilibrium. The 30 minutes turned out to be longer than needed: in each temperature step, the meander resistance became already stable within 5 to 10 minutes. For future experiments, equilibration time can be reduced, on the other hand, response time in the order of a few minutes for characterization of water samples is not a critical issue at all.
Comment 3: In 2.3, there is no discussion on how cross contamination between samples, if applicable, is eliminated when filling the channels with water/PBS/sea water assuming they are all measured using the same chip. If different chips were used, how did you ensure calibration between the various devices?
Reply: All measurements were performed with the same chip and, to avoid cross contami-nation between samples, we filled the microfluidic channel with each sample using at least 20 mL of sample. The inner volume of the microchannel was (200 ± 12.5) mL, meaning that the sample volume exceeded the channel volume by at least 100 times. Therefore, on can assume that fluid remaining from a previously studied sample was completely replaced.
The channel volume is now mentioned in Section 2.1, page 4, and the sample volumes in Section 2.3 (page 6) and in Section 3.2 (page 7), together with the argument that residues from previous measurements are completely replaced.
Comment 4: In 2.4, it is mentioned that the samples were taken from areas of intended applications in aquaculture however without a further explaination on what they are, the sample looks seemingly random as its presented. It will be beneficial to the overall understanding for the reader to understand why these samples were picked. For example why was surface water included in the sample mix and how it links to aquaculture as indicated in the title.
Reply: The surface waters were selected because of their comparatively low conductivity and our aim to calibrate the IDES sensor for a broad conductivity range. While the surface waters (# 1 – 3) are not used for aquaculture, there is a rather intensive fishing activity at these places.
In the revised version, we are putting now more emphasis on the difference between salinity and the related conductivity of the three sample categories in Table 2, see page5. As indicated in the original manuscript, certain fish species are cultivated in brackish water, where sea water and fresh water are mixed in certain ratios.
Comment 5: In 2.4, samples were stored at 4C. Does this apply to all samples and how long are they stored on average before testing? Is the sample tested at room temp so the samples will need to be brought up to room temp prior? Does this degrades the samples? Have test been done to check the difference between the results from a new sample and a stored sample? With stored samples, with the cooling and bringing it back to temp, evapouration and condensation occurs and this affects certain parameters especially in sea water which will affect conductivity. This needs to be discussed further.
Reply: The storage of all water samples at 4 oC in the refrigerator was for less than three months between sampling and the pH- and conductivity measurements. The storage had no influence on the data since several samples were measured directly after sampling at the aquaculture facility, and again after storage: the results were identical. Prior to storage, all samples were filtered with syringe filters (0.2 mm mesh) to remove microorganisms. During storage, all samples were kept in sterilized, sealed 50 mL Falcon tubes to prevent evaporation. For measuring, the samples were warmed up overnight to attain slowly the temperature of the laboratory (ca. 18.6 oC), still with sealed lids to prevent condensation from outside air humidity. Furthermore, for the pH- and conductivity measurements with the reference instrument and the IDES sensor, each sample was split into three aliquots to avoid possible cross-contamination. In the revised version, we now mention in Section 2.4, page 6, that we have indeed checked for potential pH- and conductivity alterations during storage and that a filtering, to remove microorganisms, was done before storage.
Comment 6: In 2.3, we know that conductivity measurements are strongly affected by the temperature of the sample. How is this controlled while taking measurements? What is the sample size and quantity of each sample and how many test per sample that informs your graphs.
Reply: We are not sure whether we understand the reviewer correctly. The conductivity measurements in Sections 3.1and 3.2 with the reference instrument, and the IDES sensor, took place at room temperature of 18 – 19 oC, which is indicated in the respective sections. The temperature in our lab is almost perfectly equal to the RAS facility of ILVO, which is kept constant at 18.2 oC.
However, the Reviewer asks specifically about Section 2.3 regarding the calibration of the meander-type temperature sensor, the data are shown in Section 3.4. We are aware that the impedance of an electrolyte decreases with increasing temperature, but this has no influence on the data of Figure 5 and Table 4. In turn, this means that there is indeed no parasitic current path that would shortcut the meander along the liquid atop of it. This has been added in Section 3.4 (page 11), together with an additional reference, ref. [30].
Comment 7: Statically information should contain error bars with standard deviations to representations of the variability of data and used on graphs to indicate the error or uncertainty in a reported measurement. This should extend to all plots presented in the article.
Reply: Within the original manuscript, we took already utmost care of statistics and error bars, however without mentioning this explicitly. In all figures, the error bars are smaller than symbol size, which can e.g. be seen in Fig. 5: Upon sufficient magnification, the error bars become visible within the open symbols. To clarify the situation within the revised manuscript, the following changes were made:
Table 1: we added a sentence that all data are averages of three independent measure-ments and that the uncertainties are the standard deviations, see page 4.
Table 2: it was already mentioned that the pH data are averages of three independent measurements, together with the standard deviations, see page 5.
Figure 2: all data points are averages of three independent measurements, and the standard deviations are smaller than the symbol size. This has been added to the caption of Fig. 2, see page 7.
Figures 3 a,b: The impedance spectrum of each sample was measured once (not in triplicate), however the impedance analyzer (Metrohm Autolab) registers for all data points not only the impedance amplitude, but also the width of the noise band. This noise band is at least 10-3 times smaller than the impedance amplitude, meaning again that it is smaller than the symbol sizes. A corresponding remark was added to the caption of Fig. 3, see page 8.
Figure 4: The caption has not been changed, readers will know now from Fig. 3 that the noise band around all impedance data is too small to be visualized.
Table 3: We have added two remarks to the Table caption: i) the conductivity values determined with the reference instrument are averages of three measurements, the uncertainties are the standard deviations. ii) the conductivities derived with the single-frequency algorithm from the impedance data come also with uncertainties: These were derived from the width of the noise band around the impedance values, combined with the calibration curve of Fig. 4. These additions can be found in page 10.
Figure 5: All data are the average of the meander-resistance values monitored for 10 minutes with the Hewlett-Packard multimeter; this time interval extended from 20 minutes until 30 minutes after selecting a new setpoint temperature, see also the reply to Comment 2 above. A corresponding remark has been added, and the uncertainty (noise band) is far below the size of the symbols, see page 11.
Table 4: We have added that the uncertainty of the resistance data represents the width of the noise band during a 10 minutes observation period, see page 12.
Comment 8: 3.3 should really be pulled out and discussed in a separate chapter as it discusses your method. It is kind of lost as a subsection under results.
Reply: This is a well-meant suggestion by the Reviewer, however, we need to follow the scheme of the journal with a single “Results & Discussion section”. Furthermore, Section 3.3 relies fully on the input data obtained in Section 3.2. Therefore, we did not modify the manuscript in that respect and hope that the Reviewer can agree with this.
Comment 9: Lines 422-427 and 438-442 reads identical. Only difference is the mentioned of RTD mender in 422 and just mender in 438. All other details are identical. Could the authors please clarify or make more of a distinction between the 2.
Reply: We apologize, this is a mistake from editing the original manuscript. The „original” lines 438 – 442 (in the original manuscript) have now been deleted.
Comment 10: Figure 4b way of presenting the sample impedence is somewhat confusing. Numbers not in order just looks weird unless there is a specific rationale for this way of presentation. If so this needs to be discussed.
Reply: The sample numbers in Figure 3b (we have corrected the number of this figure) are organized from top (sample # 2) to bottom (sample # 6) in the order from lowest to highest conductivity (respectively from highest to lowest impedance amplitude). The same order can be found in Figure 4 with the calibration curve that relates the impedance amplitude to the conductivity.
Based on the reviewer’s remark, we have rewritten the caption of Figure 3, see page 8.
The authors would like to thank the reviewer for the valuable comments and suggestions to improve the quality of the manuscript.
Round 2
Reviewer 1 Report
Comments and Suggestions for Authors
After evaluation, the manuscript can be accepted.
Reviewer 2 Report
Comments and Suggestions for Authors
Thank you to the authors for addressing the comments. I am happy that all my comments were addressed robustly and revisions to the manuscipt is acceptable. I am happy to recommend publication.